# Selective PPARα Modulator Pemafibrate and Sodium-Glucose Cotransporter 2 Inhibitor Tofogliflozin Combination Treatment Improved Histopathology in Experimental Mice Model of Non-Alcoholic Steatohepatitis

**DOI:** 10.3390/cells11040720

**Published:** 2022-02-18

**Authors:** Kentaro Murakami, Yusuke Sasaki, Masato Asahiyama, Wataru Yano, Toshiaki Takizawa, Wakana Kamiya, Yoshihiro Matsumura, Motonobu Anai, Tsuyoshi Osawa, Jean-Charles Fruchart, Jamila Fruchart-Najib, Hiroyuki Aburatani, Juro Sakai, Tatsuhiko Kodama, Toshiya Tanaka

**Affiliations:** 1Department of Nuclear Receptor Medicine, Laboratories for Systems Biology and Medicine (LSBM) at the Research Center for Advanced Science and Technology (RCAST), The University of Tokyo, Tokyo 153-8904, Japan; k-murakm@lsbm.org (K.M.); y-sasaki@kowa.co.jp (Y.S.); kamiya@lsbm.org (W.K.); anai@lsbm.org (M.A.); kodama@lsbm.org (T.K.); 2Pharmaceutical Division, Kowa Company, Ltd., Tokyo 189-0022, Japan; m-asahiy@kowa.co.jp (M.A.); wyano@kowa.com (W.Y.); ttakizaw@kowa.co.jp (T.T.); 3Division of Metabolic Medicine, Laboratories for Systems Biology and Medicine (LSBM) at the Research Center for Advanced Science and Technology (RCAST), The University of Tokyo, Tokyo 153-8904, Japan; matsumura-y@lsbm.org (Y.M.); jmsakai-tky@umin.ac.jp (J.S.); 4Division of Integrative Nutriomics and Oncology, Laboratories for Systems Biology and Medicine (LSBM) at the Research Center for Advanced Science and Technology (RCAST), The University of Tokyo, Tokyo 153-8904, Japan; osawa@lsbm.org; 5R3i Foundation, Picassoplatz 8, 4010 Basel, Switzerland; jean-charles.fruchart@r3i.org (J.-C.F.); jamila.fruchart@yahoo.fr (J.F.-N.); 6Genome Science Division, Laboratories for Systems Biology and Medicine (LSBM) at the Research Center for Advanced Science and Technology (RCAST), The University of Tokyo, Tokyo 153-8904, Japan; haburata-tky@umin.ac.jp; 7Tohoku University Graduate School of Medicine, Division of Molecular Physiology and Metabolism, Sendai 980-8575, Japan

**Keywords:** SPPARMα, SGLT2, ballooning, ER stress

## Abstract

Ballooning degeneration of hepatocytes is a major distinguishing histological feature of non-alcoholic steatosis (NASH) progression that can lead to cirrhosis and hepatocellular carcinoma (HCC). In this study, we evaluated the effect of the selective PPARα modulator (SPPARMα) pemafibrate (Pema) and sodium-glucose cotransporter 2 (SGLT2) inhibitor tofogliflozin (Tofo) combination treatment on pathological progression in the liver of a mouse model of NASH (STAM) at two time points (onset of NASH progression and HCC survival). At both time points, the Pema and Tofo combination treatment significantly alleviated hyperglycemia and hypertriglyceridemia. The combination treatment significantly reduced ballooning degeneration of hepatocytes. RNA-seq analysis suggested that Pema and Tofo combination treatment resulted in an increase in glyceroneogenesis, triglyceride (TG) uptake, lipolysis and liberated fatty acids re-esterification into TG, lipid droplet (LD) formation, and *Cidea/Cidec* ratio along with an increased number and reduced size and area of LDs. In addition, combination treatment reduced expression levels of endoplasmic reticulum stress-related genes (*Ire1a*, *Grp78*, *Xbp1*, and *Phlda3*). Pema and Tofo treatment significantly improved survival rates and reduced the number of tumors in the liver compared to the NASH control group. These results suggest that SPPARMα and SGLT2 inhibitor combination therapy has therapeutic potential to prevent NASH-HCC progression.

## 1. Introduction

Non-alcoholic steatosis (NASH) is a severe form of non-alcoholic fatty liver disease (NAFLD), which is closely linked to type 2 diabetes and metabolic syndrome [1,2,3]. NASH is defined as the presence of steatosis, inflammation, ballooning degeneration of hepatocytes with or without fibrosis, and the eventual development of cirrhosis and hepatocellular carcinoma (HCC). In particular, higher grades of steatosis, inflammation, and ballooning degeneration are important steps in the pathogenesis of cirrhosis and HCC and are strongly associated with morbidity and mortality of liver disease [4,5]. However, the mechanism by which lipid accumulation in hepatocytes affects NASH progression is unclear. In addition, no effective therapeutic agents have been approved for treating NASH. Therefore, the development of a therapeutic approach for NASH is urgently needed.

Lipid droplets (LDs) are storage organelles that store neutral lipids such as triglycerides (TGs) and sterol esters during excess energy states and serve as a reservoir of energy supplies during the fasting state [6,7,8]. Importantly, not only role in maintaining lipid homeostasis but also buffering function of toxic lipid species have emerged with respect to LD biology. Dysregulated LDs homeostasis is considered to induce toxic lipid release and trigger cell death through prolonged activation of signaling pathways, such as the unfolded protein response (UPR) [9,10]. However, extensive LD accumulation in hepatocytes is not always in accordance with cellular dysfunction [11,12]. Although accumulation of LDs in hepatocytes is a prerequisite step for NASH development, changes in the composition of the lipids and proteins of LDs may play an important role in the progression from NAFLD to NASH [13,14]. Thus, the investigation of LD biogenesis and degradation, as well as the regulation of hepatic fatty acid and TG metabolism by a balance of de novo lipogenesis (DNL), glyceroneogenesis, VLDL assembly and secretion, lipolysis, and fatty acid oxidation (FAO) at the transcriptional and post-transcriptional levels, is important for understanding NASH development.

Pemafibrate (Pema) is the first clinically available selective PPARα modulator (SPPARMα); it is used to improve dyslipidemia and reduce macrovascular and microvascular complications [15,16,17,18,19]. We have reported that activation of PPARα by Pema induces the expression of a series of genes involved in TG hydrolysis, fatty acid uptake, fatty acid β-oxidation, and ketogenesis in the liver, supporting its ability to reduce plasma TG [20,21]. In our previous study using STAM NASH model mice, we reported that Pema treatment prevents NASH development by reducing myeloid cell recruitment without reducing hepatic TG content [22]. Therefore, we suggest that the combination of Pema and drugs that enhance the excretion or inhibit the absorption of carbohydrates and/or lipids has the potential to alleviate LD accumulation in hepatocytes and impede NASH development.

Sodium-glucose cotransporter-2 (SGLT2) inhibitors are a class of lower blood glucose drugs that increase urinary glucose excretion by inhibiting glucose reabsorption at the proximal tubule in the kidney [23,24,25]. Recent studies suggested that SGLT2 inhibitor treatment can reduce hepatic lipid levels and alleviate NAFLD, and has blood glucose-lowering effects [26,27]. The hepatic lipid-lowering effect of SGLT2 inhibitors has been suggested, in part, based on their ability to lower circulating glucose and insulin levels, which reduces DNL. In this study, we evaluated the therapeutic potential of the combination of Pema and the SGLT2 inhibitor tofogliflozin (Tofo) in STAM NASH model mice at two time points (onset of NASH progression and HCC survival).

## 2. Materials and Methods

### 2.1. Reagents

Pema and Tofo were kindly provided by Kowa Co., Ltd. (Nagoya, Japan). Streptozotocin (STZ) was purchased from Sigma-Aldrich (St. Louis, MO, USA) and Arabic gum from Wako Pure Chemical Industries (Osaka, Japan).

### 2.2. Animal Treatment

#### 2.2.1. Progression Prevention Study

STAM mice were generated as previously described [22]. Pathogen-free pregnant C57BL/6J mice were obtained from CLEA Japan (Tokyo, Japan). All mice were housed in a temperature-controlled (24 °C) facility with a 12-h light/12-h dark cycle (08:00–20:00 h) and ad libitum access to food and water, except for the drug treatment period. Two days after birth, male mice received a subcutaneous injection of 200 μg STZ (Sigma, St. Louis, MO, USA) and were fed HFD32 (32% fat, CLEA Japan) ad libitum after 4 weeks of age of weaning. Two weeks after HFD32 feeding, mice were randomly divided into four groups: STAM control group fed HFD32 with vehicle treatment, Pema-treated group fed HFD32 with Pema (0.1 mg/kg), Tofo-treated group fed a HFD32 with Tofo (10 mg/kg), and Pema and Tofo combination (Pema 0.1 mg/kg and Tofo 10 mg/kg) for 3 weeks (6–9 weeks). Drugs were administered at 5 mL/kg body weight by oral intubation in 3% Arabic gum daily between 09:30 and 10:00 h. HFD32 was fed in a pair-feeding manner (2.3–2.8 g/mouse/day). In the drug treatment groups, animals were fed the same amount of HFD32 diet as that consumed by the control group over the preceding 24 h. In addition, normal diet (CE-2; 5% fat, CLEA Japan) fed normal group was orally administered vehicle for 3 weeks. Four hours after final administration, mice were sacrificed, and serum parameters measurement, histology, TG content determination, and gene expression analysis of liver were carried out. The study protocol was approved in accordance with the relevant guidelines and regulations of the Animal Care and Use Committee of the University of Tokyo (RAC12011, RAC170001).

#### 2.2.2. Survival Study

Male STAM mice were purchased from SMC Laboratories (Tokyo, Japan) at 5 weeks of age and were fed HFD32. C57BL/6J normal mice were purchased from Japan SLC (Shizuoka, Japan) and fed a normal diet, CE-2. All mice were housed at 23 ± 3 °C and 55 ± 15% relative humidity (RH) under a 12-h light/dark cycle (07:00–19:00 h) and provided with food and water ad libitum. At 6 weeks of age, STAM mice were divided into four groups based on body weight: control, Pema (0.00008% equivalent to 0.1 mg/kg), Tofo (0.015% equivalent to 10 mg/kg) [28,29], and Pema and Tofo combination (*n* = 20 each). C57BL/6J mice with normal chow were assigned to the normal group (*n* = 8). Pema and/or Tofo were mixed in the diet and administered to each group. The study protocol was approved in accordance with the relevant guidelines and regulations of the Animal Care and Use Committee of Tokyo New Drug Research Laboratories, Kowa Company, Ltd. (Tokyo, Japan).

### 2.3. Blood Parameter

Serum total cholesterol (TC), TG, glucose, non-esterified fatty acids (NEFA), AST, ALT, phospholipids (PL), and creatinine (CRN) levels were determined using a Labospect 003 autoanalyzer (Hitachi High-Technologies Corporation, Tokyo, Japan).

### 2.4. Histology

A histological study was performed as previously described [22]. For immunohistochemistry, blocking of endogenous peroxidase activity was performed using 0.03% H_2_O_2_ in methanol. Obtained liver sections were treated with the anti-ER-TR7 (Abcam, Cambridge, MA, USA) antibodies overnight at 4 °C. After treatment with secondary antibodies, the substrate reaction was performed using 3,3′-diaminobenzidine (Dojindo, Kumamoto, Japan) solution.

According to Kleiner et al. [30], the NAFLD activity score (NAS) was calculated. Quantitative five grades assessment of Oil Red O staining was carried out by scoring of positive areas. Quantitative estimations of ER-TR7 and Sirius-red positive areas were carried out of the positive areas in five fields. Briefly, for each animal, bright field images of stained sections were captured around the central veins at 400-fold magnification using a digital camera (DP72, Olympus, Tokyo, Japan), and quantitatively estimated using WinROOF image processing software (Mitani, Tokyo, Japan). The results were shown as the mean of five different fields in each section.

### 2.5. RNA-Sequencing

For genome-wide transcriptome analysis, RNA-sequencing (RNA-Seq) was performed as previously described [22]. Briefly, sequencing of the RNA libraries was carried out using 150-bp paired-end mode of the TruSeq Rapid PE Cluster Kit and TruSeq Rapid SBS kit (Illumina) on the Illumina HiSeq 2500 platform. RNA-seq reads were mapped onto the reference mouse genome (NCBI37/mm9) and transcriptome (UCSC gene), respectively, using Burrows-Wheeler Aligner. Transcript coordinates were converted to genomic positions, and then an optimal mapping result was chosen either via transcript or genome mapping by comparing the minimal edit distance to the reference. Local realignment was implemented within an in-house short read aligner with a smaller k-mer size (k = 11). Eventually, fragments per kilo base of exon per million fragments mapped (fpkm) values were calculated for each UCSC gene while considering strand-specific information.

### 2.6. Quantitative Real-Time PCR (qPCR)

qPCR was performed as previously described [22,31,32]. *Ppia* mRNA was used as an independent control. All primers used for qPCR are listed in Appendix A.

### 2.7. LD Analysis

LD evaluation was performed as previously described [22]. For hepatic LD analysis, “Image J” imaging software (https://imagej.nih.gov/ij/download.html (accessed on 7 August 2016) was applied. H&E staining images were opened with Image J software and converted into grayscale (8 bit). Then, the lipid drop areas were extracted by using the threshold (Min: 220, Max: 255). After eliminating blood vessels, LD areas were analyzed and quantified using the “Analyze particles” function. Quantified LD area data were firstly obtained by pixel, and then they were converted into μm^2^ (1 μm = 3 pixels, determined by scale bar size). LD diameter was also analyzed. Data were shown as the mean values from three different images of each animal. A histogram was created with Microsoft Excel spreadsheet software.

### 2.8. Statistical Analyses

All data are presented as the mean ± SEM. Homogeneity invariance was evaluated by Bartlett’s test followed by parametric or non-parametric Dunnett’s multiple comparison test (two-sided). * *p* < 0.05, ** *p* < 0.01. In the survival study, data were analyzed using the multiple log-rank test and Cox proportional hazard model.

## 3. Results

### 3.1. Pema and Tofo Combination Prevents Ballooning Degeneration of Hepatocytes

To investigate the effects of Pema and Tofo combination on NASH development in the STAM mouse model, each drug and combination was administered for three weeks. STAM mice showed significant hyperglycemia and hypertriglyceridemia; higher phospholipid, FGF21, and AST levels; lower body weight; and higher liver weight, compared to normal C57BL/6J mice (Table 1). Pema significantly reduced serum TG levels, but it did not alter serum glucose (Figure 1A,B), AST, and ALT levels. In addition, Pema significantly increased liver weight, which is a well-known effect of PPARα stimulation in rodents [33,34]. Tofo significantly reduced serum TG and glucose levels (Figure 1A,B). Pema and Tofo combination treatment effectively reduced serum TG and glucose levels and increased FGF21 levels.

H&E staining clarified that STAM control mouse liver owned liver nodules, macro- and micro-vesicular lipid accumulation, inflammatory cell infiltration, and ballooning degeneration of hepatocytes, unlike normal mouse liver (Figure 1C). Pema-treated mouse liver included less macrovesicular lipid accumulation, less ballooning degeneration, and a tendency to reduce the NAS compared to STAM control mice (Table 1). Tofo treatment reduced macrovesicular lipid accumulation and ballooning degeneration. The Pema and Tofo combination treatment significantly reduced ballooning degeneration (Figure 1D).

### 3.2. Pema and Tofo Combination Treatment Induces Lipolysis and Re-Esterification Cycles of TG in STAM Mouse Livers

Although Pema, Tofo, and combination treatment resulted in decreased macrovesicular steatosis, this reduction was not reflected in the steatosis and Oil Red O staining scores (Table 1). To verify the effect of Pema and Tofo combination treatment on STAM mouse livers, we performed a global gene expression analysis by RNA-seq using liver tissues collected from normal, STAM control, Pema-treated, Tofo-treated, and Pema and Tofo combination-treated STAM mice. We identified 125 upregulated and 68 downregulated genes in the Pema and Tofo combination treatment compared with the STAM control group according to our stringent criteria (Appendix A). These genes included almost all Pema- and/or Tofo-regulated genes. In particular, the most upregulated genes by combination treatment were PPARα target genes involved in lipid metabolism (Appendix A).

To understand the effect of Pema and Tofo combination treatment on STAM mouse livers, lipid and carbohydrate metabolism-related gene expression levels were analyzed. We found that the expression levels of genes related to TG hydrolysis, fatty acid uptake, fatty acid activation, fatty acid binding, peroxisomal and mitochondrial oxidation, and ketogenesis were increased in the STAM control group than in the normal group (Appendix A). Tofo and Pema monotherapy upregulated the expression of these genes, and the combination treatment upregulated their expression further. Importantly, the combination of Pema and Tofo dramatically increased the *Pdk4* gene expression level, indicating that it mediates the inhibition of glucose oxidation and preferential activation of FAO (Appendix A).

Increased glucose and fructose uptake in hepatocytes accelerate glycolysis and DNL to generate TG. Especially, the glycerolipid synthesis pathway (glyceroneogenesis) and the monoacylglycerol pathway are key players in TG synthesis (Figure 2A). The STAM control mouse livers exhibited higher levels of glycolysis-related gene expression than the normal mouse livers (Appendix A and Figure 2B). In addition, we found that glyceroneogenesis and re-esterification of 2-monoacylglycerol were induced in STAM control livers, in addition to simultaneous TG uptake and hydrolysis. The Pema and Tofo combination treatment did not affect glycolysis-related gene expression, but it significantly induced a series of genes involved in TG synthesis from glycerol 3-phosphate and re-esterification from monoacylglycerols and diacylglycerols generated by TG hydrolysis in STAM mouse livers (Figure 2B). These results suggest that the Pema and Tofo combination enhances the uptake and oxidation of fatty acids, TG synthesis from glycerol 3-phosphate, and the re-esterification of glycerol generated by TG hydrolysis in STAM mouse livers.

### 3.3. Pema and Tofo Combination Increases Small LDs in STAM Mouse Livers

To better understand the effect of Pema and Tofo combination treatment on steatosis in STAM mice, we measured the TG concentration in the liver (Figure 3A). The STAM control group showed a significantly increased TG content in the liver. Pema significantly increased, and Tofo and Pema and Tofo combination tended to decrease, the TG content in STAM mouse livers (Figure 3A). Because the combination of Pema and Tofo markedly improved macrovesicular steatosis (Figure 1C), we evaluated LD counts and size distributions. Pema and Tofo treatments increased the droplet number and decreased the LD area (Figure 3B,C). Furthermore, this drug combination treatment increased the percentage of cells representing small LDs (<1 μm) from 29.40% in the control to 49.65% and decreased the percentage of cells representing large LDs (>3 μm) from 39.36% in the STAM control to 7.87% (Figure 3D).

LDs consist of an inner core of neutral lipids, including TG and sterol esters, a phospholipid monolayer, and LD-associated proteins (LDAPs) [6,7,8]. Because LDAPs affect LD function and dynamics [6,7,8], we evaluated the effect of Pema and Tofo combination on LDAP expression (Figure 3E). The Pema and Tofo combination group showed increased expression of genes related to LD inner core lipid synthesis (*Agpat6*, *Dgat1*, and *Acat1*), formation (*Agpat6*, *Acsl3*, *Mettl7b*, and *Plin2*), budding (*Fitm2* and *Bscl2*), stabilization (*Plin4* and *Plin5*), lipolysis (*Pnpla2*, *Hsd17b11*, *Pcyt1a*, and *Abhd5*), expansion (*Agpat3*, *Pex3*, and *Tcp1*), and fusion (*Cidea* and *Cidec*). Among these genes, the Pema and Tofo combination induced *Cidea* expression. Recently, Sans et al. suggested that hepatic CIDEA and CIDEC correlated negatively and positively, respectively, with steatohepatitis and liver injury in mice, as well as steatosis and NASH in obese humans [35]. In addition, suppression of CIDEC has been reported to reduce LD size and stimulate lipolysis [36]. Pema, Tofo, and combination treatments induced expression of *Cidea* and *Cidec*, and combination treatment strongly induced *Cidea* gene expression, thereby increasing the *Cidea*/*Cidec* ratio. These results may contribute to the reduction in LD size and stimulation of lipolysis by combination treatment.

### 3.4. Pema and Tofo Combination Inhibits the IRE1α-XBP1-PHLDA3 Pathway

LD biogenesis and enhanced esterification of fatty acids play a key role in buffering toxic lipid species [6,7,8]. Several reports have indicated that fatty acid accumulation in hepatocytes can lead to cell dysfunction and cell death through endoplasmic reticulum (ER) stress [37,38]. UPR signaling is mainly driven by three sensors mediated by inositol requiring enzyme 1 α (IRE1α), protein kinase RNA-like endoplasmic reticulum (ER) kinase (PERK), and activating transcription factor 6 (ATF6) [39,40]. In addition, the luminal side of each UPR sensor interacts with chaperones of immunoglobulin-binding protein/glucose regulatory protein 78 (BiP/GRP78) [39,40,41]. Livers from the STAM control group showed enhanced UPR sensors (*Ire1a*, *Perk*, and *Atf6*), chaperones (*Grp78* and *Pdi1a*), antioxidant defense-regulated transcription factor (*Nrf2*), IRE1α interaction protein form apoptosis mediator complex (*Traf2*), and proapoptotic BCL-2 protein family (*Bak1* and *Bax*). Pema and Tofo combination significantly reduced *Ire1a*, *Grp78*, *Xbp1*, and *Phlda3* expression levels (Figure 4). Recently, ER stress in hepatocytes has been reported to induce PHLDA3 via the IRE1–Xbp1s pathway, which facilitates liver injury by inhibiting Akt [42]. These data suggest that the combination of Pema and Tofo prevents liver injury by inhibiting the IRE1α-XBP1-PHLD3A pathway.

### 3.5. Pema and Tofo Combination Improves HCC-Related Survival

Finally, to determine whether Pema and Tofo combination can prevent the progression of NASH to HCC, we treated STAM mice for 16 weeks. As observed in the 3-week drug treatment on NASH progression, the combination of Pema and Tofo treatment resulted in a significant decrease in serum TG and blood glucose levels (Figure 5A,B). Levels of serum AFP, an oncofetal protein that is used as a tumor marker, significantly increased in the STAM vehicle control group and decreased in the combination treatment group (Figure 5C). Kaplan-Meier survival curves show that the survival rate of the control group decreased to 10%. Pema in the diet showed a tendency to increase the survival rate (30%), and the combination of Pema and Tofo significantly improved the survival rates (50%) compared to the control group. In addition, each drug-treated group had a markedly reduced number of tumors in the liver (Figure 5E,F).

## 4. Discussion

Hepatic TG accumulation has been suggested to play a central role in NAFLD and NASH, which can progress to cirrhosis and liver failure [1,2,3,4,5,6]. The mechanisms underlying the pathogenesis of NASH in a subset of patients with steatosis have not been clarified, but several proposed hypotheses suggest that steatosis with additional factors, such as insulin resistance, oxidative stress, ER stress, and mitochondrial dysfunction, may be involved [1,2,3]. Our previous study revealed that Pema prevents NASH development without reducing the TG content in the liver [22]. We also revealed that although Pema improves macrovesicular steatosis by enhancing TG hydrolysis while simultaneously enhancing esterification of fatty acids for TG and LD biogenesis, it may not result in a sufficient TG reduction in STAM mouse livers. Based on these observations, we hypothesized that the combination of Pema with a drug that enhances the excretion of carbohydrates from the kidney via SGLT2 inhibition has the potential to improve TG accumulation and NASH development. The combination of Pema and Tofo significantly improved ballooning degeneration of hepatocytes and reduced hepatic TG accumulation. In addition, the combination of Pema and Tofo specifically reduced *Ire1a*-*Xbp1*-*Phld3a* gene expression in the NASH liver. These results suggest that the combination of SPPARMα and SGLT2 inhibitors has therapeutic potential for NASH and NASH-related HCC via reduction of ER stress-induced liver injury.

LDs are organelles that store neutral lipids, such as TG and sterol esters, as sources of energy and cell membrane synthesis [6,7,8]. When cells face excess neutral lipids, such as fatty acids and sterols, they synthesize LDs and disperse them into the cytoplasm. This process is also important for protecting cells from the toxicity associated with an excess of lipids such as fatty acids, glycerolipids, and sterols [6,7,8]. Therefore, control of LD biogenesis and consumption plays a key role in the pathogenesis of NASH. In this study, we found that the Pema and Tofo combination induced expression of genes involved in TG uptake, lipolysis, fatty acid uptake, fatty acid β-oxidation and esterification, and ketogenesis, as well as PDK4, which inhibits glucose oxidation. These results are consistent with our previous results in Pema-treated STAM mice liver [22] and were enhanced by the combination treatment used in this study. We found that Tofo improved hyperglycemia and serum TG levels and reduced TG content in the liver of STAM mice, whereas Pema reduced serum TG levels but did not reduce liver TG content. The STAM model is characterized by hyperglycemia and reduced body weight with reduced *Gck* expression, which is exclusively regulated by insulin signaling [43]. Therefore, insulin-stimulated DNL gene regulation mediated through sterol regulatory element-binding protein 1c (SREBP1c) is unlikely to contribute to these effects, suggesting that carbohydrate response element-binding protein (ChREBP) signaling regulates glycolytic and DNL genes in this model. Although PPARα activation by Pema did not affect hyperglycemia, glycolytic genes, or *G6pc* expression, which is associated with ChREBP activation by fructose, Tofo tended to reduce the expression levels of these genes. These results suggest that SGLT2 inhibitors reduce the influx of the substrate for DNL and reduce TG content in the liver of STAM mice. Our transcriptome analysis also shows that PPARα activation by Pema induced FAO and ketogenesis, but in STAM mice with a high concentration of β-hydroxybutyrate in the blood stream, it led to re-esterification of fatty acids released from the TG and sterol esters by lipolysis and uptake into the TG for LD synthesis in the liver.

In addition, we found that Pema and Tofo combination significantly increased LD number, reduced LD size, and improved macrovesicular steatosis. Consistent with the increased number of LDs, Pema and Tofo combination also induced expression of genes involved in LD inner core lipid synthesis and formation (*Agpat6*, *Dgat1*, and *Acat1*), budding (*Fitm2* and *Bscl2*), and fusion (*Cidea* and *Cidec*) proteins. Although the biological role of LD diversification has not been clarified yet, increased numbers of small LDs may explain the protection against lipotoxicity [6]. LDs have been suggested to protect against lipotoxicity under a variety of stressful conditions such as lipid overload, hypoxia, oxidative stress, autophagic flux, and dysfunctional lipolysis [6,7,8]. In fact, DGAT1-dependent LD biogenesis has been suggested to prevent lipotoxic mitochondrial dysfunction [44]. In addition, Becuwe et al. have reported that Fit2, encoded by *Fitm2*, is an evolutionarily conserved fatty acyl-CoA diphosphatase that maintains the ER structure, protects against ER stress, and enables normal lipid storage in LDs [45]. Furthermore, the differential expression of cell death-inducing DFF45-like effector (CIDE) family members CIDEC and CIDEA, recognized as regulators of LD growth, has been reported to be linked to NAFLD progression and liver injury, and CIDEA expression level decreases with NAFLD severity [35]. CIDEA and CIDEC are strongly expressed in brown adipocytes and white adipocytes, respectively, and are associated with the formation of multilocular small LDs (that are prone to lipolysis) and the storage form of unilocular LDs [36]. In this study, we found that *Cidea* was the most highly induced gene among LDAPs, and the *Cidea*/*Cidec* ratio was significantly increased by the Pema and Tofo combination. These results suggest that the combination of Pema and Tofo promotes fatty acid catabolism via lipolysis and β-oxidation while promoting re-esterification of excess fatty acids and LD biogenesis, thereby preventing lipotoxicity.

Hepatic steatosis has a risk of steatohepatitis, fibrosis, cirrhosis, liver failure, and HCC, and it was reported that dysregulation of microbial metabolites such as aromatic and branched-chain amino acid (AAA and BCAA) or of iron metabolism are related to liver fat accumulation and facilitate steatosis [46,47]; however, additional factors, such as insulin resistance, oxidative stress, ER stress, and mitochondrial dysfunction, are also involved in a disease progression [37,38]. Because the sequential esterification of fatty acids into a glycerol backbone to generate TG and budding as nascent LDs occur at the ER membrane, LDs are closely associated with ER homeostasis. Induction of UPR sensors (*Ire1a, Perk, and Atf6*), chaperones (*Grp78* and *Pdi1a*), antioxidant defense-regulated transcription factor (*Nrf2*), IRE1α interaction protein form apoptosis mediator complex (*Traf2*), and proapoptotic BCL-2 protein family (*Bak1* and *Bax*) genes were observed in the liver of STAM mice. Although numerous reports have indicated that the ER stress response plays a key role in NASH development, it is unknown which UPR sensor signaling contributes to the development of this disorder [39,40]. Pema and Tofo combination selectively reduced *Ire1a, Grp78, Xbp1*, and *Phlda3* expression levels in STAM mouse livers. Among the UPR sensors, IRE1 is the most evolutionarily conserved, implying that it plays a crucial role in cell fate determination under ER stress conditions. It has been indicated that IRE1 is capable of inducing cell fate by two distinct pathways through XBP1-mediated gene regulation and interaction with TNF receptor-associated factor 2 (TRAF2) to initiate the apoptosis signal-regulating kinase 1 (ASK1) and c-Jun N-terminal kinase (JNK) signaling cascades [48]. However, because ASK1 is activated by stress responses to ER stress, as well as ROS and TNFα, the impact of ASK1 activation by the ER stress pathway on NASH development remains unclear. A recent study reported a positive and negative effect of hepatic ASK1 ablation on NASH development in HFD-fed mice [49]. Similarly, treatment with the ASK1 inhibitor selonsertib in patients with NASH yielded controversial results [50]. However, because the effects of ASK1 inhibitors are not limited to the liver and include effects on other tissues, further studies investigating the role of ASK1 in NASH development are warranted. Recently, several reports indicated that pleckstrin homology-like domain family A member 3 (PHLDA3) functions as an AKT inhibitor and plays a crucial role in the cell fate of cancer cells [42]. In addition, PHLDA3 overexpression causes tissue injury, and the IRE1-Xbp1 pathway induces PHLDA3 overexpression, which facilitates liver injury [51]. Therefore, these results and reports suggest that the Pema and Tofo combination prevents liver injury by inhibiting the lipotoxicity-induced IRE1α-XBP1-PHLD3A pathway, thereby controlling toxic lipid esterification, LD biogenesis, and the lipolysis cycle.

Epidemiological studies have shown that NASH is closely linked to type 2 diabetes and metabolic syndrome [1,2,3]. However, the STAM mouse is recognized as a type 1 diabetes-related NASH model with hyperglycemia, reduced body weight gain, and lack of insulin secretion and fatty acid mobilization from adipose tissue. In general, storage TG in hepatocytes requires both fatty acids and glycerol and has been suggested to be mainly regulated by the pool size of fatty acid [52]. Although fat accumulation in the liver with type 1 diabetes has been reported, much less attention could be attributed to NASH prevalence of type 1 diabetes as compared to type 2 diabetes and metabolic syndrome. However, a recent report has suggested that NAFLD prevalence in patients with type 1 diabetes is considerable in meta-analysis [53], and several hypotheses have been proposed to explain the pathogenesis of liver steatosis in type 1 diabetes. These include insufficient TG secretion from the liver as VLDL, SREBP1c, and ChREBP induced DNL and conversion of sugar into fat [54]. On the other hand, the importance of circulating fatty acid influx has been suggested to contribute to increased hepatic lipid accumulation in type 2 diabetes [55], and circulating NEFA, dietary fat, and DNL have been reported to account for 59, 15, and 26% of the TG content in hepatocytes, respectively [56]. From these observations, adipose tissue-derived fatty acid influx and DNL have been suggested to play a crucial role in hepatic TG accumulation in type 2 diabetes-related NASH. In fact, it is well known that DNL is stimulated by insulin via SREBP1c activation and by influx glucose via ChREBP [57]. Insulin also activates LXRα, which in turn induces SREBP1c expression. In addition, impaired lipoprotein metabolism (VLDL export) and mitochondrial function (fatty acid entry and oxidation) have been suggested in the hepatic TG accumulation under insulin resistance [58]. Therefore, increased fatty acid influx, enhanced DNL, impaired TG secretion as VLDL, and mitochondrial dysfunction have been linked to human type 2 diabetic-related NASH. In the present study, we showed that there were no significant changes in serum NEFA in the STAM mouse compared to the normal mouse. In addition, our RNA-seq analysis indicated that impaired VLDL secretion and SREBP1c mediated DNL is unlikely to be the cause of hepatic steatosis in the STAM mouse model because VLDL assembly regulated *Mttp* was induced, insulin and SREBP1c target gene of *Gck* was reduced, and *Pck1*, which is negatively regulated by insulin, was induced. From several reports and our observations, this model may not completely reflect the human NASH liver metabolic state and may be a model in which the effect of SGLT2 inhibitors is more likely to be effective. Thus, additional studies using other NASH models with obesity and insulin resistance are warranted to evaluate the effect of the Pema and Tofo combination treatment on human NASH development.

In addition, although Pema and Tofo combination treatment significantly induced fatty acid catabolism, fatty acid re-esterification, and LD biogenesis; impeded the IRE1α-XBP1-PHLD3A pathway; and alleviated ballooning degeneration of hepatocytes, the precise underlying mechanism is still largely unknown. It is well known that hepatocytes are not equally responsible for liver metabolism, and the existence of so-called metabolic zonation based on oxygen tension has been proposed [59,60]. For example, gluconeogenesis, fatty acid β-oxidation, cholesterol synthesis, and ureagenesis are mainly considered to be performed by hepatocytes in the periportal region, where the oxygenated blood is transported via hepatic arteries, whereas glycolysis, DNL, bile acid synthesis, and xenobiotic detoxification occur in the pericentral region, which is relatively hypoxic [59,60]. Dysregulation of metabolic zonation is considered to lead to the development of lifestyle-related diseases such as obesity, diabetes, and NAFLD [61,62]. In fact, NAFLD is considered to begin with pericentral steatosis and inflammation with periportal inflammation and fibrosis considered late-occurring histological lesions [63]. However, the periportal disease has been associated with worse metabolic outcomes and more adverse hepatic fibrosis than pericentral disease [64]. In addition, interactions between hepatocytes and sinusoidal endothelial cells, Kupffer cells, and stellate cells are known to be involved in the pathogenesis of NASH [65]. Although Kupffer cells, T lymphocytes, and dendritic cells are more abundant in the periportal regions, infiltrating macrophages have been observed both in the periportal and pericentral regions [66]. Furthermore, preferential effects on the periportal and pericentral regions have been suggested for vitamin E and cysteamine, and PPARγ and FXR agonists, respectively, as per several randomized clinical trials [67,68,69]. Therefore, to understand the pathogenesis of NASH and the mechanism of therapeutic efficacy of the Pema and Tofo combination, it will be necessary to explore the spatial gene expression profile of hepatocytes and non-parenchymal cells using scRNA-seq and slide seq technologies [70,71,72].

## 5. Conclusions

In conclusion, the combination of SPPARMα and SGLT2 inhibitor treatment prevented ballooning degeneration of hepatocytes and HCC progression. Our global gene expression analysis gives evidence of the liver protective effect of the combination therapy by inhibiting the lipotoxicity-induced IRE1α-XBP1-PHLD3A pathway. Taken together with our previous report that SPPARMα treatment prevents NASH development by reducing myeloid cell recruitment without reducing hepatic TG content [22], the combination of SPPARMα and SGLT2 inhibitor presents a promising new therapy for NASH. Our results presented in this report using the NASH mouse model gives reason to hope that the combination of SPPARMα and SGLT2 inhibitor will be synergistic. Therefore, this combination is much more effective in human NASH than monotherapy and could become an ideal strategy for long-term treatment for NASH-HCC progression.

## Figures and Tables

**Figure 1 cells-11-00720-f001:**
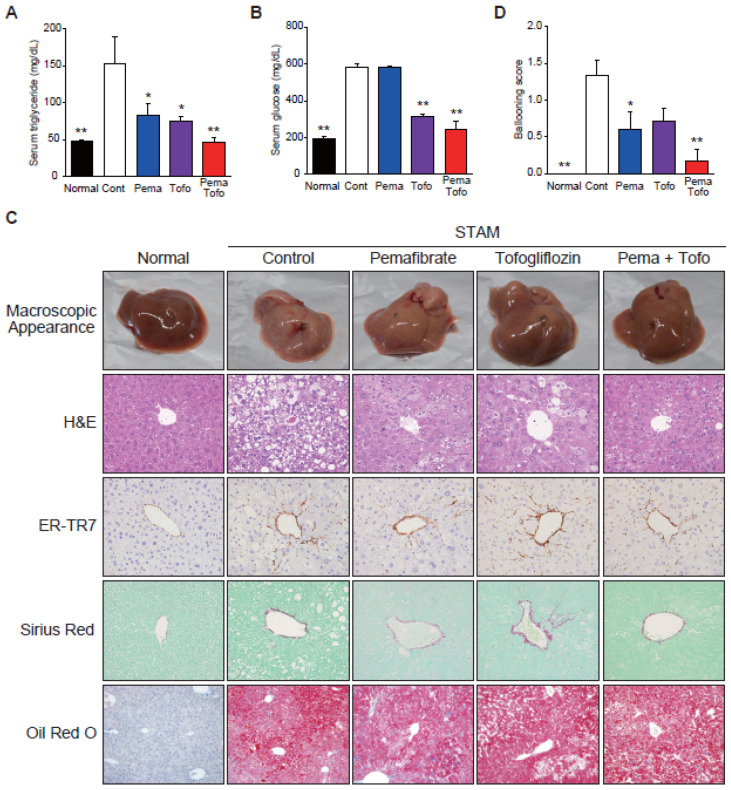
Pemafibrate and Tofogliflozin combination improves hypertriglyceridemia, hyperglycemia, macrovesicular steatosis, and ballooning score in STAM mice liver. (**A**) Serum triglyceride, (**B**) Serum glucose, (**C**) Representative gross morphology of liver, H&E stained, ER-TR7 stained, Sirius-red stained, and Oil red O stained liver section, and (**D**) Ballooning score of normal, control, pemafibrate, tofogliflozin, and pemafibrate and tofogliflozin combination-treated STAM mice. Error bars show s.e.m. * *p* < 0.05; ** *p* < 0.01: Significantly difference from STAM control group by Dunnett’s multiple comparison test.

**Figure 2 cells-11-00720-f002:**
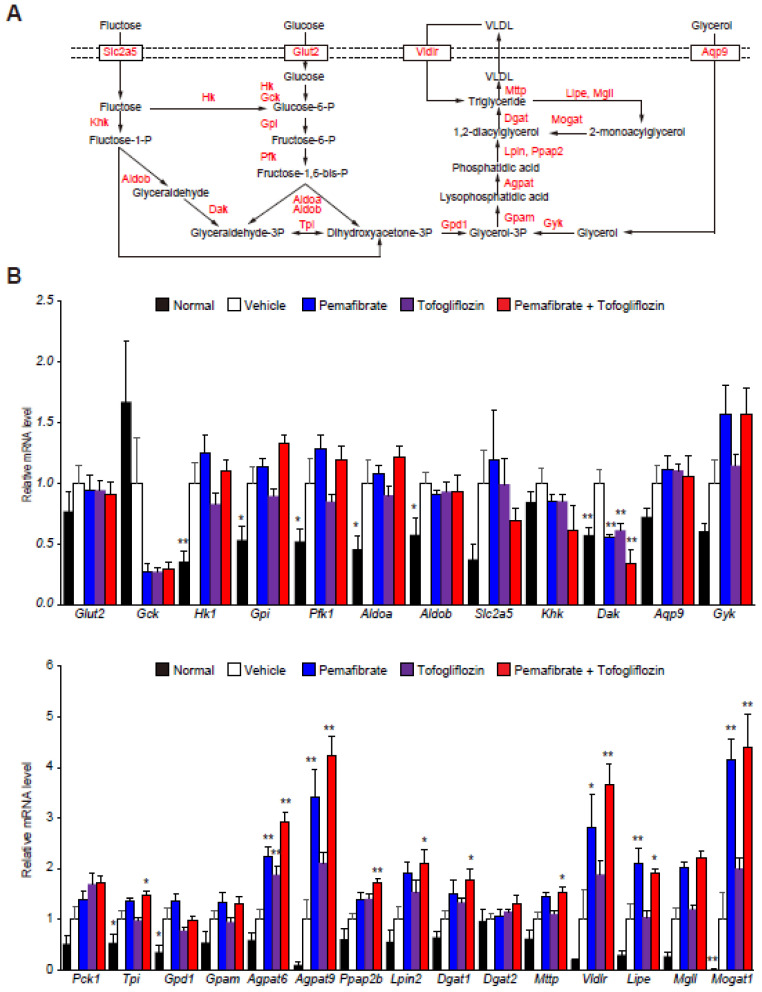
Pemafibrate and Tofogliflozin combination induces lipolysis and fatty acid re-esterification genes expression in STAM mice liver. (**A**) Schematic representation of the glycolytic and TG synthesis pathways in the liver. (**B**) qPCR validation of glycolytic and TG metabolism-related genes expression of normal, control, pemafibrate, tofogliflozin, and pemafibrate and tofogliflozin combination-treated STAM mice liver. Error bars show s.e.m. * *p* < 0.05; ** *p* < 0.01: Significantly difference from STAM control group by Dunnett’s multiple comparison test.

**Figure 3 cells-11-00720-f003:**
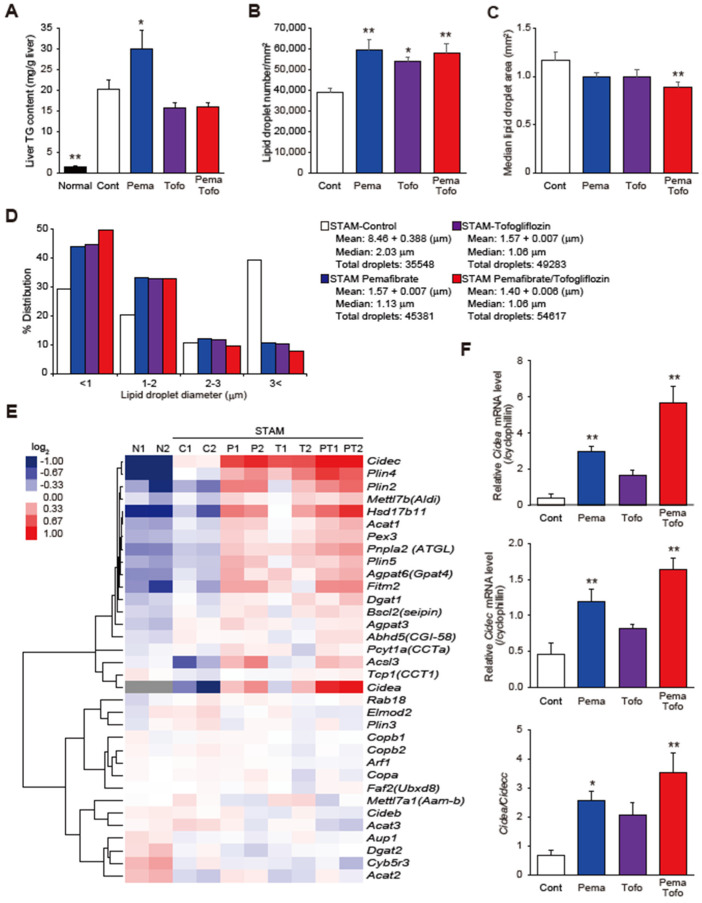
Pemafibrate and Tofogliflozin combination induces lipid droplets formation. (**A**) Liver TG content, (**B**) Lipid droplet number, (**C**) Median lipid droplet, (**D**) Lipid droplet sizes distribution, (**E**) Heatmap of hierarchical clustering of LDAPs and formation-related genes, and (**F**) *Cidea*, *Cidec*, and *Cidea*/*Cidec* ratio of control, pemafibrate, tofogliflozin, and pemafibrate and tofogliflozin combination-treated STAM mice. * *p* < 0.05; ** *p* < 0.01: Significantly difference from STAM control group by Dunnett’s multiple comparison test.

**Figure 4 cells-11-00720-f004:**
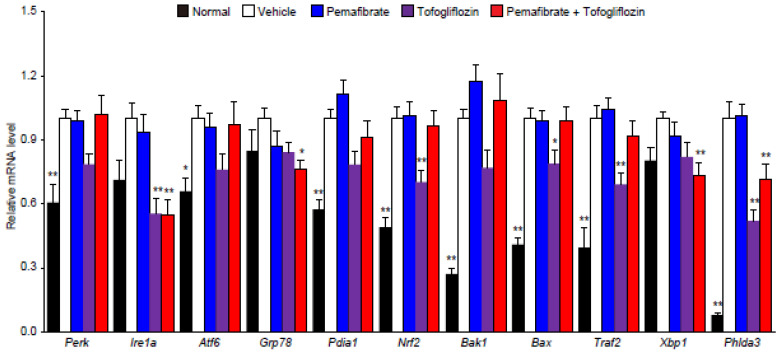
Pemafibrate and Tofogliflozin combination improves ER stress genes expression in STAM mice liver. qPCR validation of ER stress-related genes expression of normal, control, pemafibrate, tofogliflozin, and pemafibrate and tofogliflozin combination-treated STAM mice liver. * *p* < 0.05; ** *p* < 0.01: Significantly difference from STAM control group by Dunnett’s multiple comparison test.

**Figure 5 cells-11-00720-f005:**
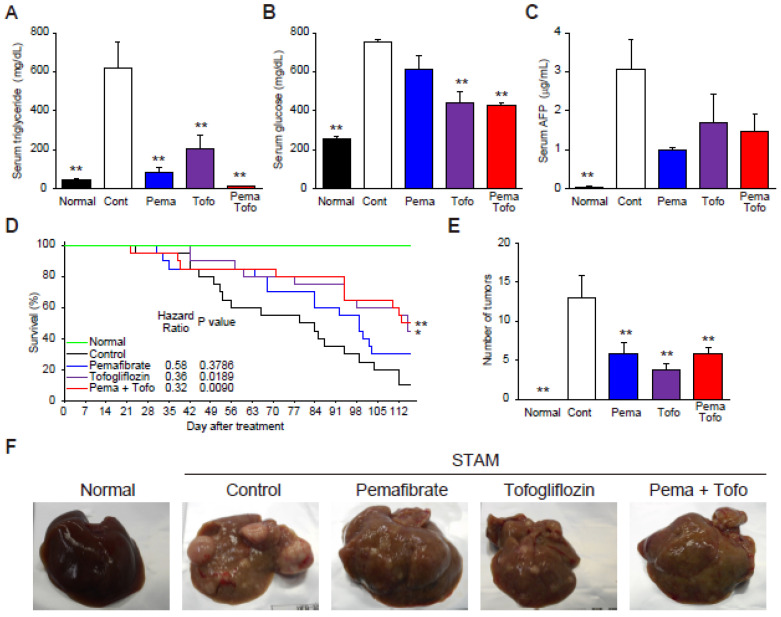
Pemafibrate and Tofogliflozin combination improves survival rate in STAM mice liver. (**A**) Serum triglyceride, (**B**) Serum glucose, (**C**) Serum AFP, (**D**) Kaplan-Meier survival curves, (**E**) Number of tumors, and (**F**) Representative gross morphology of liver from normal, control, pemafibrate, tofogliflozin, and pemafibrate and tofogliflozin combination-treated STAM mice. Log-rank *p*-value and hazard ratio were shown in the survival curve figure. Error bars show s.e.m. * *p* < 0.05; ** *p* < 0.01: Significantly difference from STAM control group by Dunnett’s multiple comparison test.

**Table 1 cells-11-00720-t001:** Effects of Pema, Tofo, and Pema and Tofo combination on body and liver weight, biochemical parameters in the serum, immunohistochemical analysis, and NAS.

	Normal	STAM			
		Vehicle	Pemafibrate	Tofogliflozin	Pemafibrate/ Tofogliflozin
n	6	6	5	7	6
Body weight (g)	23.49 ± 0.35 **	18.65 ± 0.45	18.15 ± 0.82	18.46 ± 0.49	18.25 ± 0.1
Liver weight (g)	1.31 ± 0.08 **	0.97 ± 0.03	1.43 ± 0.15 **	1.34 ± 0.03 **	1.65 ± 0.08 **
TC (mg/dL)	97.3 ± 0.7 **	171.3 ± 10.6	234.4 ± 12.1 **	194.6 ± 10.7	225.2 ± 10.7 **
PL (mg/dL)	207.7 ± 3.5 **	330.2 ± 12.9	380.4 ± 10.3 *	323.6 ± 15.7	331.3 ± 13.4
NEFA (mEq/L)	0.77 ± 0.03	0.85 ± 0.10	0.71 ± 0.06	0.91 ± 0.07	0.75 ± 0.07
β-hydroxybutylate (nmol)	17.5 ± 2.5 **	140.2 ± 8.1	149.1 ± 0.7	152.8 ± 1.6	152.7 ± 2.9
FGF21 (pg/mL)	156.1 ± 55.4 *	1579.6 ± 458.8	2449.0 ± 500.1	1430.4 ± 212.9	2759.6 ± 143.5 *
CRN (mg/dL)	0.108 ± 0.004	0.122 ± 0.005	0.100 ± 0.011	0.114 ± 0.006	0.093 ± 0.013
AST (U/L)	128.0 ± 16.4 *	181.8 ± 9.6	192.2 ± 17.3	189.6 ± 15.4	192.7 ± 10.6
ALT (U/L)	38.3 ± 7.9	71.2 ± 7.6	76.6 ± 13.2	62.9 ± 5.5	84.8 ± 12.3
Oil Red O score	0.3 ± 0.2 **	2.8 ± 0.3	3.2 ± 0.2	2.6 ± 0.2	3.2 ± 0.2
ER-TR-7 (% area)	1.667 ± 0.037 **	3.718 ± 0.451	3.074 ± 0.205	3.347 ± 0.319	3.096 ± 0.143
Sirius Red (% area)	0.257 ± 0.019 **	1.160 ± 0.205	1.096 ± 0.146	1.301 ± 0.127	1.062 ± 0.136
Steatosis	0.00 **	2.2 ± 0.3	2.2 ± 0.2	1.7 ± 0.2	2.0 ± 0.0
Inflammation	0.00 **	1.17 ± 0.17	1.20 ± 0.20	1.29 ± 0.18	1.50 ± 0.22
NAS	0.00 **	4.67 ± 0.56	4.00 ± 0.32	3.71 ± 0.36	3.67 ± 0.33

TC: total cholesterol, PL: phosphorlipids, NEFA: non-esterified fatty acid, CRN: creatinine, AST: aspartate aminotransferase, ALT: alanine aminotransferase, NAS: NAFLD activity score. Error bars show s.e.m. * *p* < 0.05; ** *p* < 0.01: Significantly difference from STAM control group by Dunnett’s multiple comparison test.

## Data Availability

The data are available upon request from the corresponding author.

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
