# Peer review of "Selective PPARα Modulator Pemafibrate and Sodium-Glucose Cotransporter 2 Inhibitor Tofogliflozin Combination Treatment Improved Histopathology in Experimental Mice Model of Non-Alcoholic Steatohepatitis"

_cells, 2022, doi:10.3390/cells11040720_

Round 1
Reviewer 1 Report
This is a nice and very interesting study evaluating the effect of pemafibrate (PPRAa) and tofoglifozin (SGLT2) combination treatment in experimental mice model of non alcoholic staetohepatitis (STAM mouse)
The authors investigated the effect of the combination treatment in many aspects of NASH:
- serum biochemical parameters (ALT, triglycerides, glucose..)
- NAFLD activity score and ballooning of hepatocytes, size and distribution of lipid droplets and liver triglycerides in liver tissue
- survival and carcinogenesis
Importantly, they have also investigated the effect on genes expression involved in lipid and glucose metabolism and also on genes related with ER stress with relatively clear and interesting results
These data elucidate some aspects of the complicated pathogenesis of NASH at the molecular level of the involved metabolic pathways and the effect of the combination treatment
However as the authors discuss this NASH model is a type 1 diabetic –related NASH model with lack of insulin secretion and reduced body weight where NASH in humans are mostly related with metabolic syndrome, obesity and insulin resistance.
So it is unknown if the same metabolic pathways apply for human with NASH.
It will be more educational to give some information in the discussion section for any existing differences in the described metabolic pathways between type I diabetes related NASH and obesity/type II diabetes related NASH
Some further minor comments
Are any PNPL3 and other related mutations involved in lipid metabolism present in this experimental model
Are they any data about Hedgehog signaling pathway activation in this model and the possible effect of the combination treatment?
Author Response
Reviewer Comments:
Reviewer 1
The authors investigated the effect of the combination treatment in many aspects of NASH:
- serum biochemical parameters (ALT, triglycerides, glucose..)
- NAFLD activity score and ballooning of hepatocytes, size and distribution of lipid droplets and liver triglycerides in liver tissue
- survival and carcinogenesis
Importantly, they have also investigated the effect on genes expression involved in lipid and glucose metabolism and also on genes related with ER stress with relatively clear and interesting results
These data elucidate some aspects of the complicated pathogenesis of NASH at the molecular level of the involved metabolic pathways and the effect of the combination treatment
However as the authors discuss this NASH model is a type 1 diabetic –related NASH model with lack of insulin secretion and reduced body weight where NASH in humans are mostly related with metabolic syndrome, obesity and insulin resistance.
So it is unknown if the same metabolic pathways apply for human with NASH.
Specific comments
It will be more educational to give some information in the discussion section for any existing differences in the described metabolic pathways between type I diabetes related NASH and obesity/type II diabetes related NASH
Reply:
We thank the reviewer for the valuable comment. Storage TG in hepatocytes requires both fatty acids and glycerol and have been suggested to mainly regulated by the pool size of fatty acid (Roden M., Nat. Clin. Pract. Endocrinol. Metab., 2, 335, 2006). Although fat accumulation in liver with type 1 diabetes has been reported, much less attention could be attributed to NASH prevalence of type 1 diabetes as compared to type 2 diabetes and metabolic syndrome. However, recent report has suggested that NAFLD prevalence in patients with type 1 diabetes is considerable in meta-analysis (de Vries M. et al., J. Clin. Endocrinol. Metab., 105, 3842, 2020). Several hypotheses have been proposed to explain the pathogenesis of liver steatosis in type 1 diabetes. These include insufficient TG secretion from liver as VLDL, SREBP1c and ChREBP induced de novo lipogenesis, and conversion of sugar into fat (Regnell SE. and Lernmark A. Rev. Diabet. Stud., 8, 454, 2011).
On the other hand, importance of circulating fatty acid influx has been suggested to contribute increased hepatic lipid accumulation in type 2 diabetes (Ravikumar B. et al. Am. J. Physiol. Endocrinol. Metab., 288: E789, 2005), and circulating NEFA, dietary fat, and de novo lipogenesis have been reported to account for ~59, 15, and 26% of the TG content in hepatocytes, respectively (Donnelly KL. et al., J. Clin. Invest., 115: 1343, 2005). From these observations, adipose tissue-derived fatty acid influx and de novo lipogenesis have been suggested play a crucial role in hepatic TG accumulation in type 2 diabetes related NASH. In fact, it is well known that de novo lipogenesis is stimulated by insulin via SREBP1c activation and by influx glucose via ChREBP (Tamura S. and Shimomura I. J. Clin. Invest., 115: 1139, 2005). Insulin also activates LXRα, which in turn induces SREBP1c expression. In addition, impaired lipoprotein metabolism (VLDL export) and mitochondrial function (fatty acid entry and oxidation) have been suggested in the hepatic TG accumulation under insulin resistance (Grefhorst A. et al., Front. Endocrinol., 11: 601627, 2021). Therefore, increased fatty acid influx, enhanced de novo lipogenesis, impaired TG secretion as VLDL, and mitochondrial dysfunction have been linked to human type 2 diabetic-related NASH.
In the present study, we showed that no significant changes in serum NEFA in STAM mouse compared to normal mouse. In addition, our RNA-seq analysis indicated that impaired VLDL secretion and SREBP1c mediated de novo lipogenesis are unlikely to be the cause of hepatic steatosis in STAM mouse model, because VLDL assembly regulated Mttp was induced, insulin and SREBP1c target gene of Gck was reduced, and Pck1, which is negatively regulated by insulin, was induced. Therefore, we speculate that both ChREBP and excess sugar regulate lipid accumulation in hepatocytes of STAM mouse.
We have made modifications for discussion section to reflect the description of differences of metabolic pathways between type I diabetes related NASH and obesity/type II diabetes related NASH.
Some further minor comments
Are any PNPL3 and other related mutations involved in lipid metabolism present in this experimental model
Are they any data about Hedgehog signaling pathway activation in this model and the possible effect of the combination treatment?
Reply to minor comments:
We agree the reviewer's comment that PNPLA3 and Hedgehog signaling pathway play critical role in the NASH development. In fact, the I148M variant of the PNPLA3 gene is well known as a risk factor for development of sever liver fibrosis (Liu YL, et al. J. Hepatol. 61, 75, 2014). Also, hedgehog pathway activation has been shown to link to NAFLD pathogenesis (Verdelho Machado M & Diehl AM. Int. J. Mol. Sci. 17, 857, 2016). In our global gene expression profile, we could not detect significant Pnpla3, hedgehog ligands (Shh, Ihh, Dhh), and Gli2 gene expression in STAM mice liver. Furthermore, significant changes in expression of Yap1, Taz, and hedgehog targets of Cxcl16 and Spp1 (osteopontin) were not observed. These results suggest that dysregulation of PNPLA3 and Hedgehog pathway may not be involved in the pathogenesis of NASH development in the STAM mouse model. However, recent report indicated that PNPLA3 variant activate Yap/Hedgehog pathways in hepatic stellate cells (HSCs) (Bruschi FV, et al. Int. J. Mol. Sci. 21, 8711, 2020). Thus, we could not exclude the possibility that our gene expression analysis did not capture the gene expression changes in HSCs. To solve these unresolved questions, we are planning the spatial gene expression profiling of hepatocytes and non-parenchymal cells using scRNA-seq and slide seq technologies.
Reviewer 2 Report
Very nice work, well presented. The reference list must be improved. I would suggest to include the following works:
1) Molecular phonemics and metagenomics of hepatic steatosis in non-diabetic obese women. Hoyles L, Fernandez-Real JM, Federici M, ET AL. Nat Med 2018 Jul; 24(7):1070-1080
2)Iron status influences non-alcoholic fatty liver disease in obesity through the gut microbiome. Mayneris-Perxachs J, Cardellini M, Hoyles L, ET AL. Microbiome 2021 May 7;9(1): 104
Author Response
Reviewer 2
Very nice work, well presented. The reference list must be improved. I would suggest to include the following works:
1) Molecular phonemics and metagenomics of hepatic steatosis in non-diabetic obese women. Hoyles L, Fernandez-Real JM, Federici M, ET AL. Nat Med 2018 Jul; 24(7):1070-1080
2)Iron status influences non-alcoholic fatty liver disease in obesity through the gut microbiome. Mayneris-Perxachs J, Cardellini M, Hoyles L, ET AL. Microbiome 2021 May 7;9(1): 104
Reply:
We thank the reviewer for the valuable comment to improve our manuscript. We have added the description about the dysregulation of microbial metabolites play a role in the fat accumulation and steatosis in the discussion section (page 13, line 408-410) and cited as reference of 47 and 48 in the revised text.